# COLLABORATIVE THRESHOLD WATERMARKING

## ABSTRACT

Consider $K$ clients who want to collaboratively train a machine learning model without sharing their data. Since each client invests considerable data and computational resources, they want the ability to verify their model's provenance by embedding a hidden signal in its weights, called a *watermark*. Clients may not trust each other and want the ability to embed a *robust* watermark that cannot be easily removed by any other client. A naive solution would be for each client to embed their own hidden watermark during training, but such a solution does not scale to many clients, as each client's contribution to the final model is bounded. We propose a trustless protocol that enables multiple clients to embed and verify a *collaborative threshold* watermark so that only a subset of $t$ or more can verify the watermark's presence, and a subset of $< t$ clients learn nothing about the watermark beyond what can be inferred from the output of the protocol. We call such a solution $(t, k)$-threshold watermarking, and it enables many clients to establish ownership with limited accuracy degradation of the model, even for large $K$. We formalize threshold watermarking and propose model watermarking schemes in the white-box setting, where the verifier can access the weights of the suspect model. We empirically demonstrate robustness against both adaptive and non-adaptive attackers on image classification tasks on multiple datasets.

## 1 INTRODUCTION

Federated learning (FL) enables multiple parties to collaboratively train large machine learning models without sharing raw data. FL has already been deployed at scale, for example, in Google Keyboard (Beaufays et al., 2019; Yang et al., 2018), Apple's voice recognition (Granqvist et al., 2020), and other privacy-sensitive applications (NVIDIA, 2021; Oldenhof et al., 2023; Muzellec et al., 2024). Recent work by Sani et al. (2024); Ye et al. (2024) envisions training large language models (LLMs) from scratch via FL to democratize model training and ownership.

Training such models consumes vast compute and data resources. However, once a model is jointly trained in FL, any client could redistribute it without the consent of any other participant. Clients must trust each other *not* to leak the model prior to engaging in the FL protocol. However, FL often operates in settings with only limited trust (e.g., without contractual obligations), meaning *trustless* mechanisms are needed to verify a model's provenance.

Model watermarking (Uchida et al., 2017; Adi et al., 2018) is a potential solution to model attribution, which embeds a hidden signal into the model that can later be detected using a secret watermarking key. A practical watermarking scheme must preserve fidelity where the scheme has negligible impact on model utility, and ensure robustness by resisting removal attempts, including surrogate model training and adaptive attacks (Boenisch, 2021; Lukas et al., 2021). Watermarking in FL introduces two additional challenges: First, if every client could independently test for the presence of the watermark, this capability can be abused to guide removal attacks. Verification should only be possible when a sufficiently large coalition of clients collaborates to verify the presence of a watermark. Second, the influence each client has on the model during training is bounded and diminishes as the number of clients grows. This makes an approach where each client embeds their own watermark infeasible at scale. A trustless protocol is needed where $K$ clients can collaboratively embed a shared watermark, but only $t \leq K$ clients can verify the watermark.

To address these challenges, we introduce *(t,k)-threshold model watermarking*. This solution enables $k$ clients to collaboratively embed a shared watermark into a model, while ensuring that fewer than

Figure 1: A high-level overview of threshold watermarking, where clients use shares of secret watermarking keys during and after training to collaboratively embed and verify a watermark.

$t$ clients learn nothing about the watermark beyond what can be inferred from the protocol's output. The guarantee is that at least $t$ clients must collaborate to detect the presence of a watermark in a *suspect* model. We demonstrate across multiple image classification datasets, such as CIFAR-10, CIFAR-100, and TinyImageNet, that our method scales to many users (up to $k = 128$), is *robust* against adaptive removal attacks, and has a small impact on the model's utility.

## 1.1 Contributions.

Our contributions can be summarized as follows:

- We propose the first $(t, k)$-threshold watermark for FL that scales to many clients (large $k$) and ensures that only $\geq t$ clients can collectively verify the presence of a watermark.
- We empirically show that our watermark has a negligible impact on the model's accuracy.
- Our watermark is robust against removal attacks, including adaptive attackers who know our method is being used and can access intermediate model checkpoints during training.

## 2 Background

### 2.1 Federated Learning (FL)

FL enables multiple clients to collaboratively train a shared model without exchanging raw data. Each client $k$ has a dataset of size $n_k$, and the global model $\theta$ is trained to minimize the overall weighted loss

$$F(\theta) = \frac{1}{n} \sum_k n_k F_k(\theta), \tag{1}$$

where $F_k(\theta)$ is the local loss and $n = \sum_k n_k$. Training proceeds iteratively: (1) The server sends the current model to clients, (2) clients update it using their local data, and (3) the server aggregates these updates, (e.g., by weighted averaging). Secure aggregation (Bonawitz et al., 2017) ensures that the server sees only the combined updates and not individual gradients.

### 2.2 Model Watermarking

A model watermark is a hidden signal that can be extracted with a secret watermarking key given access to the model. Formally, watermarking schemes are defined by the following procedures:

- $\tau \leftarrow \text{KEYGEN}()$: A randomized function that generates a secret watermarking key $\tau$.
- $\theta_w \leftarrow \text{EMBED}(\theta, \tau)$: Given a model's parameters $\theta$ and a secret key $\tau$, this function returns watermarked parameters $\theta_w$ such that the watermark is embedded in the model.
- $y \leftarrow \text{VERIFY}(\theta_s, \tau)$: This function attempts to detect the watermark in the suspect model $\theta_s$ using the key $\tau$, and outputs a p-value to reject the null hypothesis that the detected signal is present due to random chance.

We use the *same* secret key $\tau$ for both the embedding and verification algorithms and ensure that only the key holder can successfully verify and claim ownership of the watermark. A *white-box* watermark requires that VERIFY has access to the model's parameters for verification, whereas a black-box watermark requires only API access. In this work, we focus on white-box watermarks.

### 2.3 CRYPTOGRAPHIC PREMISES

#### 2.3.1 COMMITMENT SCHEMES

A *commitment scheme* (Brassard et al., 1988; Katz & Lindell, 2020) is a two-phase protocol between a sender and receiver. It consists of a *commit* algorithm to fix a value $m$ with randomness $r$, producing a commitment $\mathcal{C}$, and an *open* algorithm where the sender reveals $(m, r)$ for verification. It has two characteristics: (i) **Hiding:** Commitments reveal nothing about $m$. (ii) **Binding:** It is infeasible to open the same commitment to two different values[1].

#### 2.3.2 SECRET SHARING

A *secret sharing scheme* (Stinson, 2005; Katz & Lindell, 2020) defines a dealer who holds a secret and wants to distribute it to $n$ clients so that only any subset of at least $t$ clients can reconstruct it.

**Shamir's threshold scheme.** In the classical $(t, n)$-threshold scheme (Shamir, 1979), the secret is embedded as the constant term of a random degree-$(t-1)$ polynomial $P(x)$ over a finite field $\mathbb{F}_q$. Each participant $i$ receives a share $s_i = P(x_i)$ and any $t$ shares uniquely interpolate $P(x)$, while fewer than $t$ reveal nothing[2].

**Additive secret sharing.** In the $n$-party additive scheme (Escudero, 2022), a dealer samples $n-1$ shares $a_1, \ldots, a_{n-1}$ uniformly at random from a field $\mathbb{F}_q$ and sets:

$$a_n = a - \sum_{i=1}^{n-1} a_i \pmod{q} \tag{2}$$

The secret is reconstructed by summing all shares modulo $q$, This scheme achieves perfect secrecy where any subset of $n-1$ or fewer shares reveals no information about the secret $a$.

## 3 THREAT MODEL

We consider a standard FL setup in a trustless setup: $K$ clients want to train a joint model without revealing their local data. Each client contributes (i) data and (ii) compute. At any stage in the FL training process, the clients want the ability to collectively verify the presence of a watermark if at least $t \leq K$ clients collaborate. For example, clients may want to retain ownership or deter unauthorized usage of the model. We assume a trustless environment where clients only trust a set of at least $t$ of other clients. A security challenge is that fewer than $t$ clients should not learn anything about the watermark to prevent unauthorized removal of the watermark.

**Defender's Capabilities and Goal.** We consider each client to be a 'defender'. We assume clients have secret local data that other clients cannot access or infer. The composition of clients remains fixed throughout the FL training process, i.e., no client drops or is replaced. Our security assumption is that clients are *honest-but-curious*: they follow the FL protocol correctly and do not try to corrupt the watermark's embedding during training, but will try to learn as much as possible about the watermark from intermediate outputs. There is no trusted server, meaning that model updates are securely aggregated (e.g., via the protocol by Bonawitz et al. (2017)), which ensures that individual client updates are encrypted and only their aggregated form is accessible to the server. This prevents the server or any individual client from learning individual gradients computed on local data.

We further assume the presence of a trusted dealer that, at the outset of training, *obliviously distributes secret shares* to all users. The dealer does not participate in the FL protocol and their presence is not required during verification. During training, we assume each user is honest-but-curious,

---

[1]Formal definitions are given in Appendix B.1

[2]Detailed constructions are given in Appendix B.2

but after training, we assume that up to $t - 1$ of the $K$ users may collude and attempt to remove the watermark. We provide two variants of our protocol, where one leverages the presence of a trusted server to act as a dealer and cut down the communication rounds, and the other assumes that the server is not trusted, thus clients embed the watermark via additive shares. The defender's objective is to design a robust, privacy-preserving watermarking mechanism that supports collaborative verification and resists tampering or removal by less than $t$ subsets of clients after training.

**Adversary's Capabilities and Goals.** Our adversary has access to all intermediate model checkpoints produced throughout the FL process. In addition, they have access to a larger subset of the total training data used to train the model than all other clients, which they may use to fine-tune the model post-training. Importantly, the adversary does not possess enough data to independently train a high-utility model from scratch. Their objective is to obtain a model with comparable accuracy to the original but without the presence of the watermark signal.

## 4   CONCEPTUAL APPROACH

We now formally introduce threshold watermarking, an algorithm to embed white-box verifiable model watermarks, and describe adaptive attacks against our method.

**Definition 1** (Threshold Watermarking). *A $(t, k)$-threshold white-box watermarking scheme for FL involves $K$ clients collaboratively training a model $\theta$. It consists of multiple algorithms such that:*

1. Setup*: Takes a parameter c, threshold $t$, number of clients $K$ and outputs a secret watermarking key $\tau \in R^d$ where $d$ is the dimensions of $\theta$, a public commitment $C$ to $\tau$, and $K$ secret shares of $\tau$. Each client $k$ receives a share.*

2. Embed*: Takes the current model $\theta_r$, client gradients $\{\nabla f_k(\theta_r)\}_{k=1}^K$ and client shares $\{s_k\}_{k=1}^K$. Outputs an updated model $\theta_{r+1}$ where the watermark $\tau$ is embedded via perturbations derived from the shares.*

3. Verify*: Takes a suspect model $\theta_s$, a set of $\geq t$ client shares $\{s_j\}_{j \in \mathcal{S}, |\mathcal{S}| \geq t}$ and the public commitment $C$. Outputs 1 if the watermark is present and 0 otherwise.*

*The scheme must satisfy*

1. **Correctness***: For a model $\theta_w$ trained using our scheme, the watermark can be verified by any subset $\mathcal{S}$ of clients with $|\mathcal{S}| \geq t$.*

2. **Soundness (Binding)***: It is computationally infeasible for an adversary to produce a model $\theta'$ and claim it contains a watermark $\tau'$ (where $\tau' \neq \tau$) such that* Verify *outputs 1 using the legitimate shares and commitment $C$.*

3. **Threshold Security***: For any coalition of $m < t$ malicious clients, it is computationally infeasible to (i) accurately reconstruct the full secret key $\tau$ and (ii) modify a watermarked model $\theta_w$ to $\theta'$ such that for $|\mathcal{S}| \geq t$ without substantially degrading model utility.*

### 4.1   WATERMARK CONSTRUCTION

**Intuition.** We exploit the high dimensionality of model parameters to embed a watermark without noticeably affecting model performance. Each client adds a small perturbation along their watermark share, scaled by the magnitude of their local gradients, so that the changes are proportional to the client's natural updates. Over many clients and rounds, these tiny perturbations accumulate along the watermark vector $\tau$, creating a detectable pattern in the model's parameter space. During verification, we use cosine similarity to measure alignment between the suspect model and the watermark vector; even though each perturbation is small, the high-dimensional space ensures that the signal stands out statistically, allowing reliable detection while preserving the model's utility. A similar construction in the centralized case (without FL) has been explored by Uchida et al. (2017).

### 4.1.1 WATERMARK SETUP

Suppose $K$ clients participate in FL training with global model $\theta_r$ at round $r$. In the presence of a trusted dealer that samples a watermarking key $\tau$ that creates a commitment $\mathcal{C}$ for $\tau$ and publishes it to all K clients, for each client $k$ has a local dataset $\mathcal{D}_k$, receives a watermark share $s_k$ from Shamir Secret Sharing of $\tau$.

**Trustless server.** In scenarios where the server cannot be trusted to act as the dealer, we instead use additive secret sharing to embed the watermark. We denote this by

$$\{a_k\}_{k=1}^K \;\leftarrow\; \text{AdditiveShare}(\tau). \tag{3}$$

Concretely, the dealer samples $K-1$ random values and defines the final share to satisfy the additive property:

$$a_1, a_2, \ldots, a_{K-1} \xleftarrow{i.i.d} \mathcal{N}(0,1), \quad a_K = \tau - \sum_{k=1}^{K-1} a_k. \tag{4}$$

Each client $k$ receives its corresponding share $a_k$, which is used to embed the watermark, where the watermark key is later reconstructed implicitly during secure aggregation of client updates. Note that, unlike in Eq. 2, no modulus operation is required here since secure aggregation itself ensures secrecy of the shares.

### 4.1.2 WATERMARK EMBEDDING

Each client computes local model updates $\nabla\theta_r^{(k)} := \theta_r^{(k)} - \theta_{r-1}$ through standard FL, and maintains an exponential moving average (ema) of update norms for adaptive watermark scaling.

Let $\text{ema}_k$ be the gradient magnitude tracker for client $k$, $c$ be a scaling constant, and $\text{scale}_{\text{total}}$ be the aggregated scaling factor across all clients. The EMA tracker is updated as:

$$\text{ema}_k^{(r)} = \beta \cdot \text{ema}_k^{(r-1)} + (1 - \beta) \cdot \|\nabla\theta_r^{(k)}\|_2 \tag{5}$$

By maintaining $\text{ema}_k$, we can scale each client's watermark perturbation according to their recent update magnitudes. Larger EMA values correspond to larger model updates, which can tolerate stronger watermark perturbations without significantly degrading model quality. Conversely, smaller EMA values indicate smaller updates, requiring weaker watermark perturbations to preserve model fidelity.

At each training round, the watermark perturbation scaling is computed as:

$$\text{scale}_k = \|\nabla\theta_r^{(k)}\|_2 \cdot c \cdot \text{ema}_k^{(r)} \tag{6}$$

where $\|\nabla\theta_r^k\|_2$ is the norm of client $k$'s model update, and $\text{ema}_k$ tracks historical update magnitudes. One can track update magnitudes via this exponential moving average to ensure adaptive watermark strength.

**Trusted Server.** The server receives clean models $\{\theta_r^{(k)}\}_{k=1}^K$ and scaling factors $\{\text{scale}_k\}_{k=1}^K$, computes the total scaling $\text{scale}_{\text{total}} = \sum_{k=1}^K \text{scale}_k$, and produces:

$$\theta_r = \frac{1}{K}\sum_{k=1}^K \left(\theta_r^{(k)} + \text{scale}_k \cdot \tau\right) \tag{7}$$

**Trustless Server.** Clients use secure aggregation to compute $\text{scale}_{\text{total}}$, embed their watermark shares locally, and send watermarked models:

$$\theta_r = \frac{1}{K}\sum_{k=1}^K \underbrace{(\theta_r^{(k)} + \text{scale}_{\text{total}} \cdot a_k)}_{\text{from client } k} \tag{8}$$

In both cases, we obtain an identical aggregated model:

$$\theta_r = \frac{1}{K}\sum_{k=1}^K \theta_r^{(k)} + \frac{\text{scale}_{\text{total}}}{K}\tau \tag{9}$$

Therefore, the global model obtained by locally embedding additive watermark shares under secure aggregation (trustless server) is identical to the model obtained by applying the watermark centrally on a trusted server. Refer to Appendix C for more details.

### 4.1.3 WATERMARK VERIFICATION

To verify the presence of the watermark, at least $t$ clients collaboratively reconstruct the watermark key $\tau$ from their secret shares $\{s_i\}_{i \in \mathcal{S}}$ using polynomial interpolation, where $|\mathcal{S}| \geq t$.

Given a suspect model $\theta_s$, we compute the cosine similarity between $\theta_s$ and $\tau$ and calculate the $z$-score:

$$z = \frac{\cos(\theta_s, \tau) - \mu}{\sigma}, \tag{10}$$

where $\mu$ and $\sigma$ denote the mean and standard deviation of cosine similarities between unwatermarked models and random vectors.

The model $\theta_s$ is considered watermarked if $|z| \geq z^*$ for a preset threshold $z^*$ (e.g., $z^* = 4$). Note that the adversary learns nothing about the watermark beyond what can be inferred from the protocol's output (the $z$-score). We refer to Appendix D for an implementation of our threshold watermarking protocol in both trusted and trustless settings.

## 5 EXPERIMENTS

### 5.1 EMPIRICAL VALIDATION OF NORMALITY ASSUMPTION

To validate the assumption of normality in the cosine similarity distribution (introduced in Section 4.1.3), We trained five independent ResNet-18 models for each dataset. For each model $\theta_i$, we computed cosine similarities with 2000 random vectors. We then aggregated the similarities across models to obtain combined distributions from which $\mu$ and $\sigma$ were estimated.

As shown in Figures 2 and 3, individual model distributions exhibit consistent behavior, and the combined distributions closely match fitted normal curves. Q-Q plots confirm that the empirical cosine similarities align well with the estimated normal distribution. These results empirically support the use of the normality assumption in our statistical test for watermark verification.

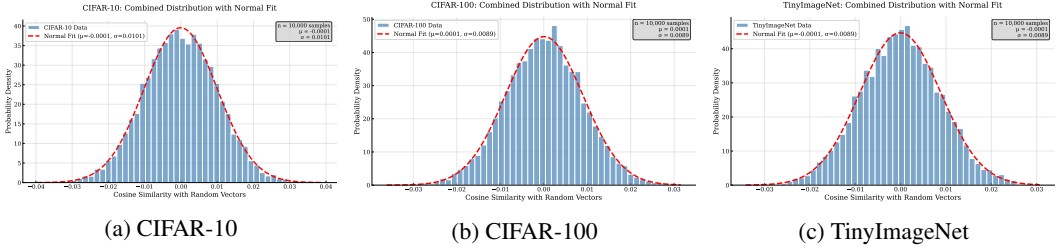

(a) CIFAR-10  (b) CIFAR-100  (c) TinyImageNet

Figure 2: Combined cosine similarity distributions for different datasets on ResNet18 models, with fitted normal distributions.

### 5.2 EXPERIMENTAL SETUP

**Training.** Experiments were conducted on the CIFAR-10, CIFAR-100 datasets (Krizhevsky, 2009) and Tiny ImageNet (Le & Yang, 2015). We use the ResNet-18 architecture (He et al., 2016) and train it on A6000 GPUs for all of our experiments. All models were randomly initialized before training. To evaluate model utility, we report top-1 accuracy. We quantify robustness by measuring the $z$-score as described in Section 4.1.3. For exact implementation details, please refer to Appendix E.

**Baseline.** We compare our method with a naive baseline, where each client embeds their own watermark into the model weights during FL training.

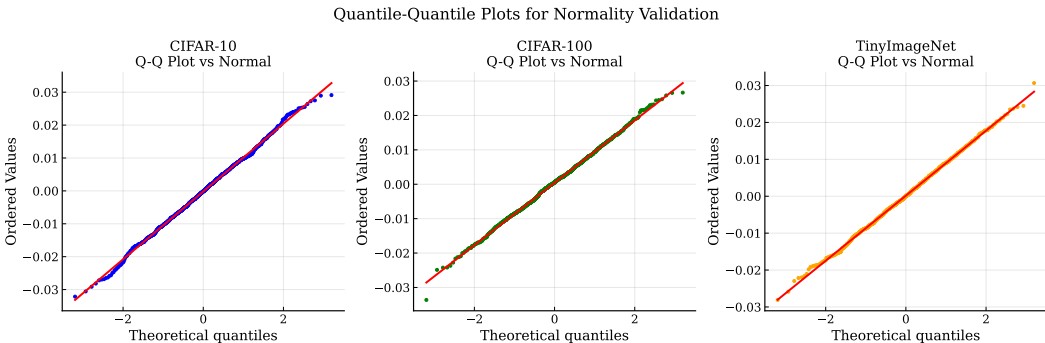

Figure 3: Q-Q plots comparing observed cosine similarity quantiles against theoretical normal distribution quantiles. Points following the red diagonal line indicate a good fit to the normal distribution.

## 5.3 WATERMARK SCALABILITY

Figure 4a illustrates the effect of scaling the number of clients on the watermark signal strength for both the baseline and our proposed method. We observe that for the baseline, each client embeds a unique signal, which causes the overall signal to weaken as the number of clients increases. Notably, when $K \geq 16$, the z-score not statistically significant and below our threshold. In contrast, our collaborative watermarking approach supports scaling to larger $K$ (demonstrated up to $K = 128$) even with a smaller scaling factor ($c = 0.025$, compared to $c = 0.1$ in the baseline). Interestingly, the z-scores increase in our method when keeping the watermark embedding strength $c$ constant as the number of clients increases.

Figure 4b shows that the statistical significance with which our watermark can be detected increases predictably with the watermark strength hyperparameter $c$ (scaling factor). As $c$ increases, the corresponding $z$-scores rise consistently across all datasets. This indicates that our scheme is both scalable and tunable: the watermarking strength can be directly controlled by adjusting $c$. In the next section, we further investigate the trade-off between watermark strength and model accuracy.

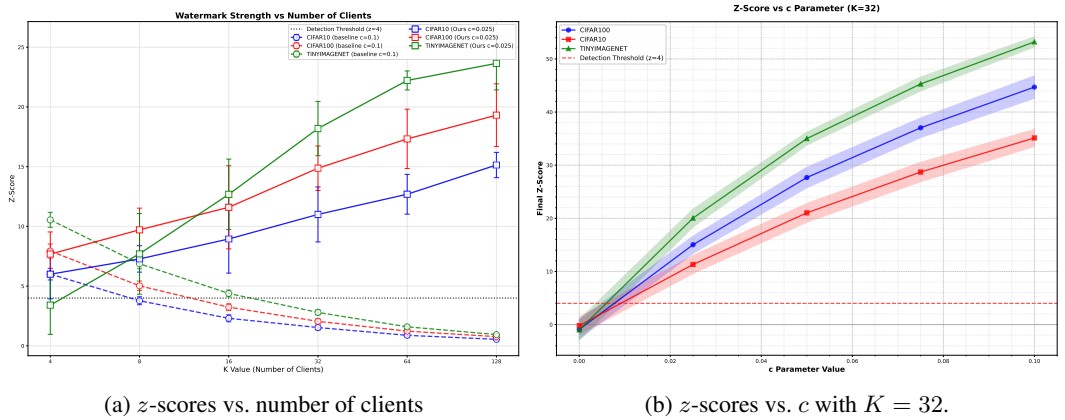

(a) $z$-scores vs. number of clients      (b) $z$-scores vs. $c$ with $K = 32$.

Figure 4: Watermark scalability. (a) Our method sustains statistically significant $z$-scores up to $K = 128$, whereas the baseline collapses beyond $K = 16$. (b) Increasing the scaling factor $c$ consistently boosts $z$-scores across datasets, showing that watermark strength is tunable.

## 5.4 WATERMARK FIDELITY

Table 1 shows the model's accuracy relative to the watermark embedding strength $c$. At low watermark strengths, the impact on accuracy is small, e.g., at $c = 0.025$, accuracy decreases on CIFAR-10 by only 0.12%, on CIFAR-100 by 0.21%, and on TinyImageNet by 0.25%. Increasing $c$ to 0.05 or 0.075 results in slightly larger but still modest reductions: CIFAR-10 decreases by 0.28% and

0.64%, CIFAR-100 by 1.15% and 2.13%, and TinyImageNet by 0.47% and 0.57%, respectively. This demonstrates that even at moderate strengths, the model maintains high accuracy.

At higher watermark strengths, e.g., $c = 0.1$, accuracy noticeably degrades: 1.1% for CIFAR-10, 4.6% for CIFAR-100, and 2.5% for TinyImageNet. We find that less complex datasets like CIFAR-10 are less sensitive in terms of impact on the test accuracy.

Table 1: Mean test accuracy (%) across datasets for different watermark strengths $c$ ($K = 32$).

| Watermark Strength | CIFAR-10 | CIFAR-100 | TinyImageNet |
|---|---|---|---|
| No Watermark | 88.11 | 61.61 | 54.21 |
| 0.025 | 87.99 | 61.40 | 53.96 |
| 0.050 | 87.83 | 60.46 | 53.74 |
| 0.075 | 87.47 | 59.48 | 53.64 |
| 0.100 | 87.12 | 58.75 | 52.87 |

## 5.5 WATERMARK ROBUSTNESS

We experiment with the following watermark removal attacks.

1. **Fine-Tuning:** The attacker locally fine-tunes the watermarked model.

2. **Adaptive Fine-Tuning:** The attacker uses intermediate global model checkpoints throughout training to estimate the watermarking key. Then the attacker fine-tunes the model adaptively to increase the distance to the predicted key.

3. **Quantization:** The attacker post-processes the model by quantizing its weights.

4. **Pruning:** The attacker resets parameters by magnitude using either global unstructured pruning across layers or structured pruning of output channels.

5. **Distillation:** The adversary trains a student network to imitate a watermarked teacher.

Figure 5 shows the robustness results on CIFAR-100 ($K = 32$, $c = 0.025$) via Pareto frontiers. We observe that even with larger attack budgets (up to 20% of the data) or structural modifications such as 90% pruning, watermark $z$-scores remain above the detection threshold ($z = 4$). Adaptive fine-tuning yields the best results for the attacker, but it still cannot erase the watermark without substantial accuracy degradation. The only attack that reliably removes our watermark is distillation, which is to be expected since we use a white-box watermark. However, distillation requires (i) a high computational effort to re-train the model and (ii) substantial amounts of training data. Our experiments confirm that our watermark withstands diverse black-box and adaptive attacks, and an attacker is only successful if (i) they can substantially degrade the model's utility unless or (ii) with access to substantial compute or data resources. We refer to Appendix F for more details.

## 6 DISCUSSION & RELATED WORK

**Centralized Training Model Watermarking.** In traditional centralized machine learning, Model watermarking has been extensively studied to protect the intellectual property of the model owner by embedding a secret watermark that can later be extracted to prove ownership of the model Boenisch (2021); Lukas et al. (2021). White-box Watermarking methods directly modify the model's parameters to embed the watermark (Lv et al., 2023; Darvish Rouhani et al., 2019; Wang & Kerschbaum, 2021; Wang et al., 2020). This might involve altering the distribution of weights or encoding a specific pattern within them; such methods require access to the model's internal parameters for verification. On the contrary, black-box Watermarking does not require access to the model's internal structure. Instead, they rely on training the model to behave in a specific way when presented with trigger inputs (Adi et al., 2018; Jia et al., 2021; Bansal et al., 2022; Yang et al., 2023b). The watermark is verified by observing the model's output on these triggers. These methods involve a single source of randomness; however, in FL settings, Each of the clients and server needs to contribute to the randomness without being able to remove others' randomness.

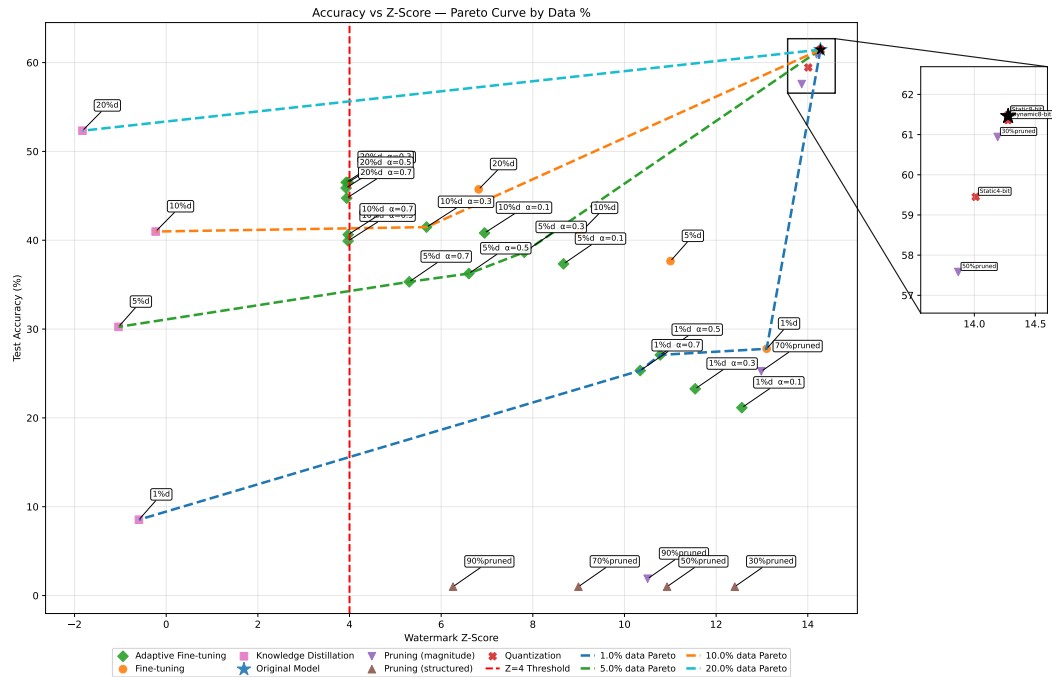

Figure 5: Robustness analysis on CIFAR-100 with $K = 32$ and $c = 0.025$. We report the trade-off between task accuracy and watermark $z$-score under five attack types: (i) adaptive fine-tuning, (ii) plain fine-tuning, (iii) knowledge distillation, (iv) pruning (magnitude and structured), and (v) quantization. The original model is shown as a blue star. Dashed curves denote Pareto frontiers for 1%, 5%, 10%, and 20% of the training data, while the red dashed line marks the detection threshold ($z = 4$).

**Federated Learning Model Watermarking** FL introduces complexities to watermarking due to its distributed nature. Key considerations include who embeds the watermark and how to maintain its persistence throughout training rounds. Despite having fewer works on FL settings, several watermarking schemes have emerged (Lansari et al., 2023):

**Server-side Embedding**: Where the central server embeds the watermark, often by retraining with a trigger set after each aggregation round (Tekgul et al., 2021; Shao et al., 2025; Li et al., 2022). This approach requires trust in the server.

**Client Embedding**: One or more clients embed watermarks, adding trigger sets to their local data (Li et al., 2023; Yang et al., 2023a; Liu et al., 2021; Chen et al., 2024; Liang & Wang, 2023; Xu et al., 2024). This does not require server trust but requires mechanisms to manage multiple client watermarks without overwriting other clients' randomness.

## 7 CONCLUSION

We introduced a $(t, K)$-threshold watermarking scheme for federated learning that enables collaborative model ownership without requiring a trusted server. Our construction guarantees that only coalitions of at least $t$ clients can verify provenance, while smaller groups learn nothing beyond the protocol's output. We instantiated our protocol in the *white-box setting* and showed empirically that it is robust to watermark removal, including adaptive attackers. We demonstrate that threshold watermarking is both practical and scalable to large $K$, making it a viable solution for collaborative ownership of machine learning models. More broadly, we hope that our method benefits many users when training large models and enable attribution through collaborative watermarking.

## 8 Reproducibility Statement

We are committed to ensuring the reproducibility of our results. To this end, we provide all necessary details regarding the experimental setup, datasets, and evaluation within Appendix E. We also share source code and instructions to facilitate replication of our findings. Any external datasets or tools used are publicly available and properly referenced. Our aim is to enable independent researchers to reproduce and verify the results reported in this study.

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

# A    NOTATION TABLE

Table 2: Notation table for collaborative threshold watermarking.

| Symbol | Definition |
|---|---|
| $K$ | Total number of clients in federated learning (FL). |
| $k$ | Index of a client, $k \in \{1, \dots, K\}$. |
| $D_k$ | Local dataset of client $k$. |
| $n_k$ | Number of samples in $D_k$. |
| $n$ | Total number of samples, $n = \sum_{k=1}^{K} n_k$. |
| $F_k(\theta)$ | Local loss function of client $k$. |
| $F(\theta)$ | Global objective, weighted sum of local losses. |
| $\theta_r$ | Global model parameters at round $r$. |
| $\theta_0$ | Initial global model. |
| $\theta_w$ | Final watermarked global model after $T$ rounds. |
| $\theta_r^{(k)}$ | Locally trained model of client $k$ at round $r$. |
| $\nabla\theta_r^{(k)}$ | Local model update of client $k$ at round $r$. |
| $\theta_s$ | Suspect model under verification. |
| $\tau$ | Secret watermarking key vector. |
| $\tau'$ | Estimated key (attack approximation of $\tau$). |
| $\mathcal{C}$ | Public commitment to $\tau$ (from dealer). |
| $s_i$ | Shamir secret share of $\tau$ held by client $i$. |
| $a_i$ | Additive share of $\tau$ (trustless-server setting). |
| $c$ | Scaling constant for watermark strength. |
| $\beta$ | EMA decay factor for update magnitudes. |
| $\text{ema}_k$ | Exponential moving average tracker for client $k$. |
| $\text{scale}_k$ | Scaling factor for client $k$'s watermark perturbation. |
| $\text{scale}_{\text{total}}$ | Aggregated sum watermark scaling factor. |
| $z$ | Standardized verification statistic. |
| $z^*$ | Detection threshold for watermark verification. |
| $\mu, \sigma$ | Mean and standard deviation of cosine similarities, estimated from unwatermarked models. |
| $E$ | Local training epochs per round. |
| $T$ | Number of global FL rounds. |
| $\eta$ | Learning rate. |
| $z_s, z_t$ | Student and teacher logits (distillation attack). |
| $T_{\text{distill}}$ | Temperature parameter for distillation softening. |
| $\alpha$ | Mixing factor (attack or distillation trade-off parameter). |
| $p\%$ | Fraction of training data available to adversary during attacks. |

# B    CRYPTOGRAPHIC PRELIMINARIES

## B.1    COMMITMENT SCHEMES

A *commitment scheme* (Brassard et al., 1988; Katz & Lindell, 2020) is a cryptographic protocol consisting of two phases: the *commitment phase* and the *opening phase*. It is executed between two parties, a sender (committer) and a receiver, and is designed to satisfy two fundamental security properties: *hiding* and *binding*.

In the *commitment phase*, the sender commits to a value $m$ in such a way that it cannot change the value later (*binding*), while ensuring that the receiver learns nothing about the committed value until it is revealed in the *opening phase* (*hiding*). Formally, a commitment scheme is defined as a tuple of algorithms (`Commit`, `Verify`), where:

1. **Commit:** The `Commit` algorithm is a probabilistic algorithm that takes as input a message $m \in \mathcal{M}$, chosen from a pre-defined message space $\mathcal{M}$, and a random string $r$, chosen from a randomness space $\mathcal{R}$. It outputs a commitment $\mathcal{C}$:

$$\mathcal{C} \leftarrow \texttt{Commit}(m, r) \tag{11}$$

   The commitment $c$ is sent to the receiver as an unchangeable representation of $m$.

2. **Open:** In the *opening phase*, the sender reveals the committed message $m$ and the randomness $r$ to the receiver. The receiver verifies the validity of the commitment by executing a deterministic verification algorithm, `Verify`, which takes as input the commitment $\mathcal{C}$, the message $m$, and the randomness $r$, and outputs a binary decision:

$$\texttt{Verify}(\mathcal{C}, m, r) \rightarrow \{0, 1\} \tag{12}$$

   where the output 1 indicates that the verification is successful (i.e., the commitment is consistent with $m$ and $r$), and 0 indicates failure.

A commitment scheme is secure if it satisfies both the hiding and binding properties :

1. **Hiding:** A commitment scheme satisfies the *hiding property* if for all probabilistic polynomial-time (PPT) adversaries $\mathcal{A}$, the advantage of $\mathcal{A}$ in distinguishing commitments to any two messages $m_0, m_1 \in \mathcal{M}$ is negligible. Formally, for all PPT adversaries $\mathcal{A}$ and for all $m_0, m_1 \in \mathcal{M}$:

$$\left| \Pr[\mathcal{A}(\texttt{Commit}(m_0, r)) = 1] - \Pr[\mathcal{A}(\texttt{Commit}(m_1, r)) = 1] \right| \leq \mathrm{negl}(\lambda) \tag{13}$$

   where $r \xleftarrow{\$} \mathcal{R}$ is chosen uniformly at random, $\lambda$ is the security parameter, and $\mathrm{negl}(\lambda)$ denotes a negligible function in $\lambda$.

2. **Binding:** A commitment scheme satisfies the *binding property* if it is computationally infeasible for any PPT adversary $\mathcal{A}$ to produce two distinct openings $(m_1, r_1)$ and $(m_2, r_2)$ such that $m_1 \neq m_2$ but they produce the same commitment. Formally, for all PPT adversaries $\mathcal{A}$:

$$\Pr\left[ \begin{matrix} (m_1, r_1, m_2, r_2) \leftarrow \mathcal{A}(1^\lambda): \\ m_1 \neq m_2 \wedge \texttt{Commit}(m_1, r_1) = \texttt{Commit}(m_2, r_2) \end{matrix} \right] \leq \mathrm{negl}(\lambda) \tag{14}$$

   where $\lambda$ is the security parameter and $\mathrm{negl}(\lambda)$ denotes a negligible function in $\lambda$.

## B.2 SECRET SHARING

A *secret sharing scheme* (Stinson, 2005; Katz & Lindell, 2020) enables a dealer to divide a secret among $n$ participants so that only certain groups (usually of size at least $t$) can reconstruct the secret, while smaller groups learn nothing.

Shamir's scheme (Shamir, 1979) is a $(t, n)$-threshold scheme based on polynomial interpolation over a finite field $\mathbb{F}_q$, where $q$ is a prime number such that $q > \max(s, n)$. The scheme is defined as follows:

1. **Share Generation** The dealer selects a random polynomial $P(x)$ of degree $t - 1$ over $\mathbb{F}_q$:

$$P(x) = s + a_1 x + a_2 x^2 + \cdots + a_{t-1} x^{t-1},$$

   where $s \in \mathbb{F}_q$ is the secret, and $a_1, \ldots, a_{t-1} \in \mathbb{F}_q$ are chosen uniformly at random. The shares are computed as:

$$s_i = P(x_i), \quad \text{for } i = 1, 2, \ldots, n,$$

   where $x_1, \ldots, x_n$ are distinct, nonzero elements of $\mathbb{F}_q$. Each participant receives a pair $(x_i, s_i)$.

2. **Reconstruction:** To reconstruct the secret $s$, any subset of $t$ shares $(x_{i_1}, s_{i_1}), \ldots, (x_{i_t}, s_{i_t})$ is used. The secret is recovered as:

$$s = P(0) = \sum_{k=1}^{t} s_{i_k} \cdot \ell_k(0) \tag{15}$$

where $\ell_k(x)$ is the Lagrange basis polynomial defined as:

$$\ell_k(x) = \prod_{\substack{j=1 \\ j \neq k}}^{t} \frac{x - x_{i_j}}{x_{i_k} - x_{i_j}} \mod q \tag{16}$$

Shamir's scheme satisfies the following properties:

1. **Confidentiality:** Any subset of fewer than $t$ shares reveals no information about $s$.

2. **Recoverability:** Any subset of $t$ or more shares uniquely determines the polynomial $P(x)$, consequently the reconstruction of $s$.

## C  EQUIVALENCE OF TRUSTED AND TRUSTLESS AGGREGATION

*Proof.* We show that the global model updates produced by the **Trusted Server** case (Eq. 7) and the **Trustless Server** case (Eq. 8) are mathematically equivalent.

**Properties.**

1. Additive shares satisfy $\sum_{k=1}^{K} a_k = \tau$.

2. The scaling factors aggregate as $\text{scale}_{\text{total}} = \sum_{k=1}^{K} \text{scale}_k$.

**Trusted Server (Eq. 7):**

$$\theta_r = \frac{1}{K} \sum_{k=1}^{K} \left( \theta_r^{(k)} + \text{scale}_k \cdot \tau \right)$$

$$= \frac{1}{K} \sum_{k=1}^{K} \theta_r^{(k)} + \frac{\text{scale}_{\text{total}}}{K} \tau.$$

**Trustless Server (Eq. 8):**

$$\theta_r = \frac{1}{K} \sum_{k=1}^{K} \left( \theta_r^{(k)} + \text{scale}_{\text{total}} \cdot a_k \right)$$

$$= \frac{1}{K} \sum_{k=1}^{K} \theta_r^{(k)} + \frac{\text{scale}_{\text{total}}}{K} \sum_{k=1}^{K} a_k$$

$$= \frac{1}{K} \sum_{k=1}^{K} \theta_r^{(k)} + \frac{\text{scale}_{\text{total}}}{K} \tau.$$

**Conclusion.** Both aggregation strategies yield the same global model:

$$\theta_r = \frac{1}{K} \sum_{k=1}^{K} \theta_r^{(k)} + \frac{\text{scale}_{\text{total}}}{K} \tau.$$

Thus, the trusted and trustless procedures yield identical global updates. $\square$

# D    DETAILS OF THRESHOLD WATERMARKING PROTOCOL

---

**Algorithm 1** Threshold Watermarking Protocol (trusted server setting vs. trustless server setting).

---

**Require:** $K$ clients, threshold $t$, rounds $T$, epochs $E$, learning rate $\eta$, watermark scale $c$, EMA decay factor $\beta$

1: **Setup Phase:**
2: Dealer samples watermark vectors $\tau \sim \mathcal{N}(0, I)$ and nonce $r \sim \mathcal{U}$
3: Dealer publishes commitment $\mathcal{C} = \text{Commit}(\tau, r)$
4: Dealer creates Shamir shares: $\{s_k\}_{k=1}^{K} \leftarrow \text{ShamirShare}(\tau, t, K)$
5: Dealer sends share $s_k$ to each client k
6: **if** Dealer != Server (Trustless) **then**
7:     Dealer creates $\{a_k\}_{k=1}^{K} \leftarrow \text{AdditiveShare}(\tau)$
8:     Dealer sends $a_k$ shares to each client k
9:     Dealer destroys original watermark $\tau$
10: **end if**
11: Initialize global model $\theta_0$ and EMA trackers $\text{ema}_k \leftarrow 0$ for all clients

12: **for** round $r = 1$ to $T$ **do**
13:     Server broadcasts $\theta_{r-1}$ to all clients
14:     **Client Local Training:**
15:     **for** each client $k$ in parallel **do**
16:         $\theta_k^{(0)} \leftarrow \theta_{r-1}$
17:         **for** epoch $e = 1$ to $E$ **do**
18:             **for** batch $\mathcal{B}$ in $\mathcal{D}_k$ **do**
19:                 $\theta_r^{(k)(e)} \leftarrow \theta_r^{(k)(e-1)} - \eta \cdot \nabla_\theta \mathcal{L}(\theta_r^{(k)(e-1)}, \mathcal{B})$
20:             **end for**
21:         **end for**
22:         $\text{ema}_k \leftarrow \beta \cdot \text{ema}_k + (1 - \beta) \cdot \|\nabla \theta_r^{(k)}\|_2$
23:         $\text{scale}_k \leftarrow \|\nabla \theta_r^{(k)}\|_2 \cdot c \cdot \text{ema}_k$
24:         Send trained model $\theta_r^{(k)}$ to server
25:     **end for**
26:     **Server Aggregation:**
27:     **if** Dealer == Server (Trusted Server) **then**
28:         Server receives trained models $\{\theta_r^{(k)}\}_{k=1}^{K}$ and scaling factors $\{\text{scale}_k\}_{k=1}^{K}$
29:         Server computes $\text{scale}_{\text{total}} \leftarrow \sum_{k=1}^{K} \text{scale}_k$
30:         $\theta_r \leftarrow \frac{1}{K} \sum_{k=1}^{K} \theta_r^{(k)} + \text{scale}_{\text{total}} \cdot \tau$
31:     **else**
32:         Server aggregates $\text{scale}_{\text{total}} \leftarrow \sum_{k=1}^{K} \text{scale}_k$ using secure aggregation protocol (e.g.,(Bonawitz et al., 2017)) and broadcasts to clients
33:         **for** each client $k$ **do**
34:             $\theta_r^{(k)'} \leftarrow \theta_r^{(k)} + \text{scale}_{\text{total}} \cdot a_k$
35:             Send watermarked model $\theta_r^{(k)'}$ to server
36:         **end for**
37:         $\theta_r \leftarrow \frac{1}{K} \sum_{k=1}^{K} \theta_r^{(k)'}$
38:     **end if**
39: **end for**

40: **Verification:**
41: Coalition $|\mathcal{S}| \geq t$ reconstructs: $\tau \leftarrow \text{ShamirReconstruct}(\{s_i\}_{i \in \mathcal{S}})$
42: Verify commitment: $\text{Verify}(\mathcal{C}, \tau, r) = \text{True}$
43: Compute verification score: $z \leftarrow \frac{\cos(\theta_s, \tau) - \mu}{\sigma}$
44: **return**  verification score $z$

---

## E  IMPLEMENTATION DETAILS

**Datasets.** We conduct experiments on CIFAR-10, CIFAR-100 (Krizhevsky, 2009), and Tiny ImageNet (Le & Yang, 2015). Since CIFAR-10 and CIFAR-100 do not provide validation sets, we split the original training set into 80% for training and 20% for validation. For evaluation, we select the model checkpoint with the highest validation accuracy. For CIFAR-10 and CIFAR-100, we apply the AutoAugment policy designed for CIFAR-10. For Tiny ImageNet, training images are augmented using random horizontal flips ($p = 0.5$), random rotations ($\pm15°$), and color jittering (brightness, contrast, and saturation up to 0.4; hue up to 0.1).

**Model.** All experiments use the ResNet-18 architecture (He et al., 2016), with models initialized randomly before training. Implementations and training are carried out in PyTorch (Paszke et al., 2019).

**Training.** In our FL setup, we vary the number of clients from 4 to 128 while keeping the global batch size fixed at 2048, ensuring an equal number of batches across experiments. We use the AdamW optimizer (Loshchilov & Hutter, 2019) with a learning rate of $1 \times 10^{-3}$, weight decay of $1 \times 10^{-4}$, and betas $(0.9, 0.999)$. Training runs for 300 rounds with a single epoch per round. The EMA decay factor $\beta$ for our watermark is set to 0.9. To ensure reproducibility and reduce the risk of seed overfitting, all experiments are repeated with three random seeds: 0, 1, and 2.

**Hardware.** All training and post-training attack experiments are conducted on a single NVIDIA A100 GPU with 40 GB of VRAM.

## F  ROBUSTNESS ANALYSIS

This section details the robustness evaluation of our watermarking method. We describe the attacks used to showcase our watermark robustness, along with the experimental setup, so that our results can be reproduced and extended by other researchers. Robustness is assessed under a range of attack scenarios, with measurements reported in terms of (i) classification accuracy on the test set and (ii) watermark detectability via the standardized $z$-score.

Unless otherwise stated, all experiments are conducted on CIFAR-100 watermarked models trained with client counts $K = 32$, $c = 0.025$.

### F.1  FINE-TUNING

Fine-tuning is the most direct strategy an attacker can attempt, simply retrain the released model on a subset of data in the hope of diminishing the watermark signal.

The attack is assumed to have access to only a subset of the training data, with size $p\% \in \{1, 5, 10, 20\}$ of the full training dataset. Each subset is sampled uniformly at random from the training set with fixed seeds to ensure reproducibility. Fine-tuning proceeds for 100 epochs using the AdamW optimizer with learning rate $1 \times 10^{-3}$, weight decay $1 \times 10^{-4}$, betas $(0.9, 0.999)$, and batch size 128.

Fine-tuning gradually reduces the alignment between model parameters and watermark flip vectors, but does not cross the decision threshold within 100 epochs; on the other hand, the model utility is diminished by a very high margin even with 20% of the available data, as shown in Figure 6.

### F.2  ADAPTIVE FINE-TUNING

We assume that a participant who participated in the training and knows that the algorithm in use tries to remove the watermark, leaving the knowledge gained through the training. One way of achieving this is to estimate the bias by accumulating the gradient over the training process and use it as an **estimated key** $\tau'$ to remove the watermark. The attack can be formulated as an optimization problem to minimize the following loss function :

$$\mathcal{L}' = (1 - \alpha)\mathcal{L} + \alpha \|\tau'\|_1 \tag{17}$$

where $\mathcal{L}$ is the model's utility loss and $\alpha$ controls the tradeoff between the utility loss function and the removal objective. When $\alpha = 0$, this reduces to standard fine-tuning. We evaluate $\alpha \in \{0.1, 0.3, 0.5, 0.7\}$. Other hyperparameters (epochs, optimizer, batch size) match fine-tuning. For each epoch, we log accuracy and $z$-score. Pareto frontiers are constructed by retaining non-dominated points in the $(\text{accuracy}, z)$ plane.

Optimizing to minimize the signal of the estimated key improves the fine-tuning attack effectiveness. Increasing $\alpha$ drives $z$ down more rapidly, but once $z$ falls below the detection threshold ($z < 4$), the model also exhibits very high accuracy degradation.

### F.3 QUANTIZATION

Quantization discretizes model weights to a lower numerical precision. Quantization threatens embedded signals because fine-grained correlations in parameter space may be destroyed by discretization.

We apply symmetric weight-only quantization to all Conv/Linear layers, leaving biases and BatchNorm parameters unchanged. Three settings are considered:

- **Static8:** per-tensor symmetric 8-bit quantization,
- **Static4:** per-tensor symmetric 4-bit quantization,
- **Dynamic8:** per-output-channel symmetric 8-bit quantization.

For per-tensor quantization, a single scale is computed as the maximum absolute value in the tensor divided by the representable integer range. For per-channel quantization, scales are computed independently for each output channel.

Results shown in Figure 8, all quantized models still carry the watermark signal despite the quantization method used.

### F.4 PRUNING

Pruning reduces model size by eliminating parameters. Both unstructured (sparsity-inducing) and structured (channel-removal) pruning can disrupt embedded watermarks by discarding weights that carry watermark information. We evaluate two pruning strategies:

- **Magnitude pruning:** global unstructured pruning across all Conv/Linear weights, removing a fraction with the smallest magnitude.
- **Structured pruning:** per-layer pruning of entire output channels, removing a fraction of channels per module.

Pruning ratios $\in \{0.3, 0.5, 0.7, 0.9\}$ are tested. Models are evaluated immediately after pruning without retraining. Structured pruning is implemented with $\ell_1$-norm channel selection.

Both pruning strategies reduce the watermark signal as the pruning ratio increases, but never cross the decision threshold as shown in Figure 9.

### F.5 KNOWLEDGE DISTILLATION

An attacker can attempt to distill the model by training a student network to imitate the watermarked teacher.

The student is trained using a mixture of ground-truth supervision and distillation from teacher logits. Given temperature $T = 3.0$ and weighting $\alpha = 0.5$, the loss is

$$\mathcal{L} = \alpha \, \text{KL}\left(\sigma\left(\tfrac{z_s}{T}\right) \| \sigma\left(\tfrac{z_t}{T}\right)\right) + (1 - \alpha) \, \text{CE}(z_s, y) \tag{18}$$

where $z_s$ and $z_t$ denote the student and teacher logits.

The student is trained for 100 epochs with Adam (learning rate $1 \times 10^{-3}$, batch size 128) on subsets $p\% \in \{1, 5, 10, 20\}$ of the exported training data. We report test accuracy, transfer rate, and final watermark $z$-score.

Distillation can produce student models that approach the teacher's task accuracy while noticeably attenuating the watermark signal, since the functional behavior is transferred without preserving parameter-level correlations. However, we observe that distillation doesn't produce a high utility model when the available data fraction is below 20%, illustrated in Figure 10, meaning that a substantial portion of the training data is required for the attack to succeed. Moreover, training a new student network is computationally expensive, making this approach significantly more costly than lightweight post-processing attacks such as pruning or quantization.

## G  LIMITATIONS AND FUTURE WORK

While our work introduces threshold watermarking as a novel direction for collaborative model provenance, we emphasize several limitations. First, our scheme focuses exclusively on the *white-box setting*, where the verifier has direct access to model parameters. Although this is a natural starting point, it limits applicability in scenarios where only black-box access is possible. Extending threshold watermarking to the black-box setting remains an important and challenging open problem.

We consider this work an initial step toward the broader goal of enabling $(t, K)$-threshold watermarking in collaborative learning. Additionally, we envision extending collaborative watermarking to content such as images and text within collaborative inference scenarios, allowing multiple providers to jointly prove ownership.

## H  LLM USAGE DECLARATION

We acknowledge the use of large language models (LLMs) to assist in refining portions of the text in this manuscript. Their use was strictly limited to improving readability, grammar, spelling, and style. All ideas, interpretations, and conclusions presented in this work are solely the responsibility of the authors.

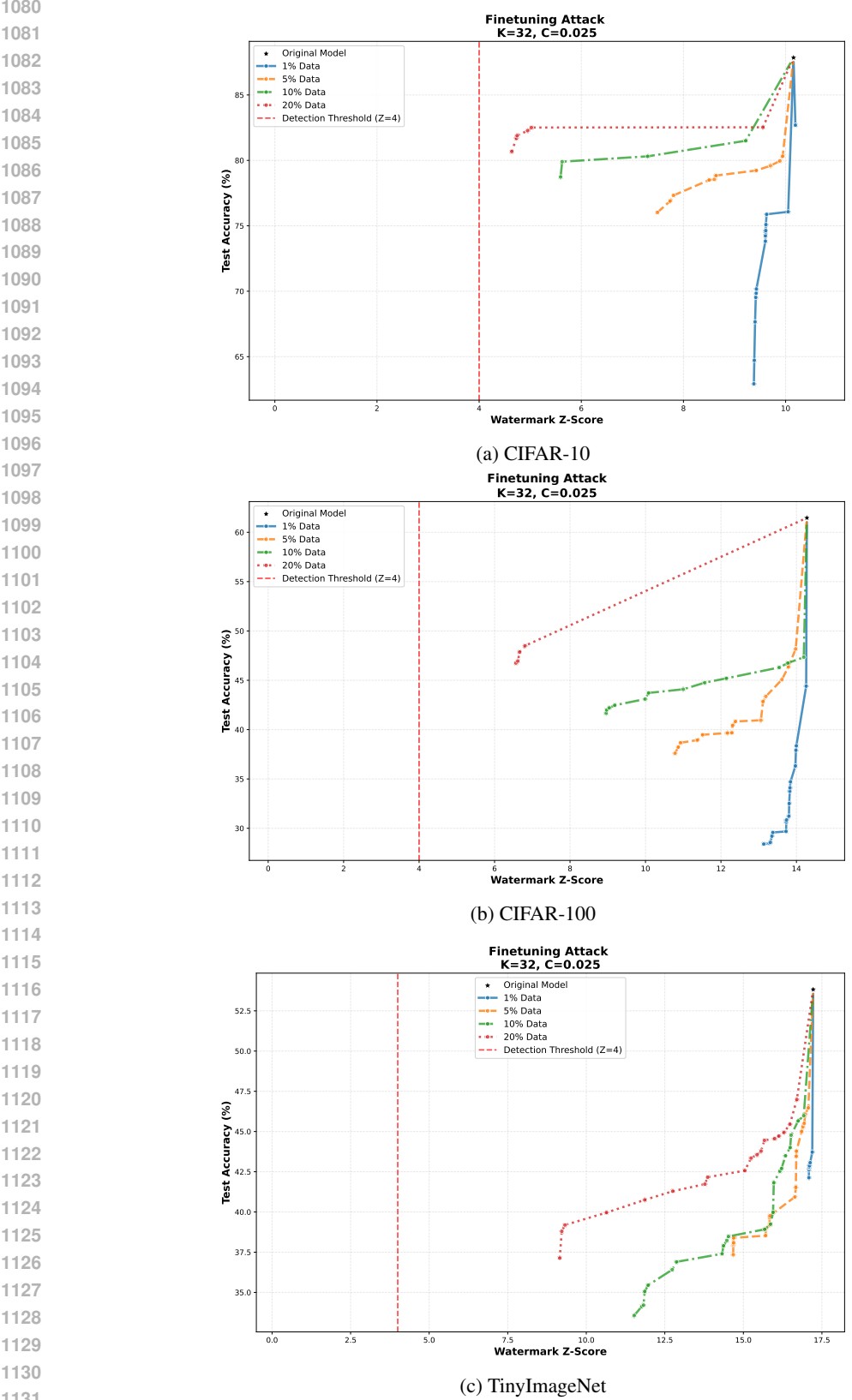

(a) CIFAR-10

(b) CIFAR-100

(c) TinyImageNet

Figure 6: Fine-tuning attack: test accuracy versus watermark $z$-score under different fractions of fine-tuning data (5%, 10%, 20%). The dashed red line denotes the detection threshold ($z = 4.0$).

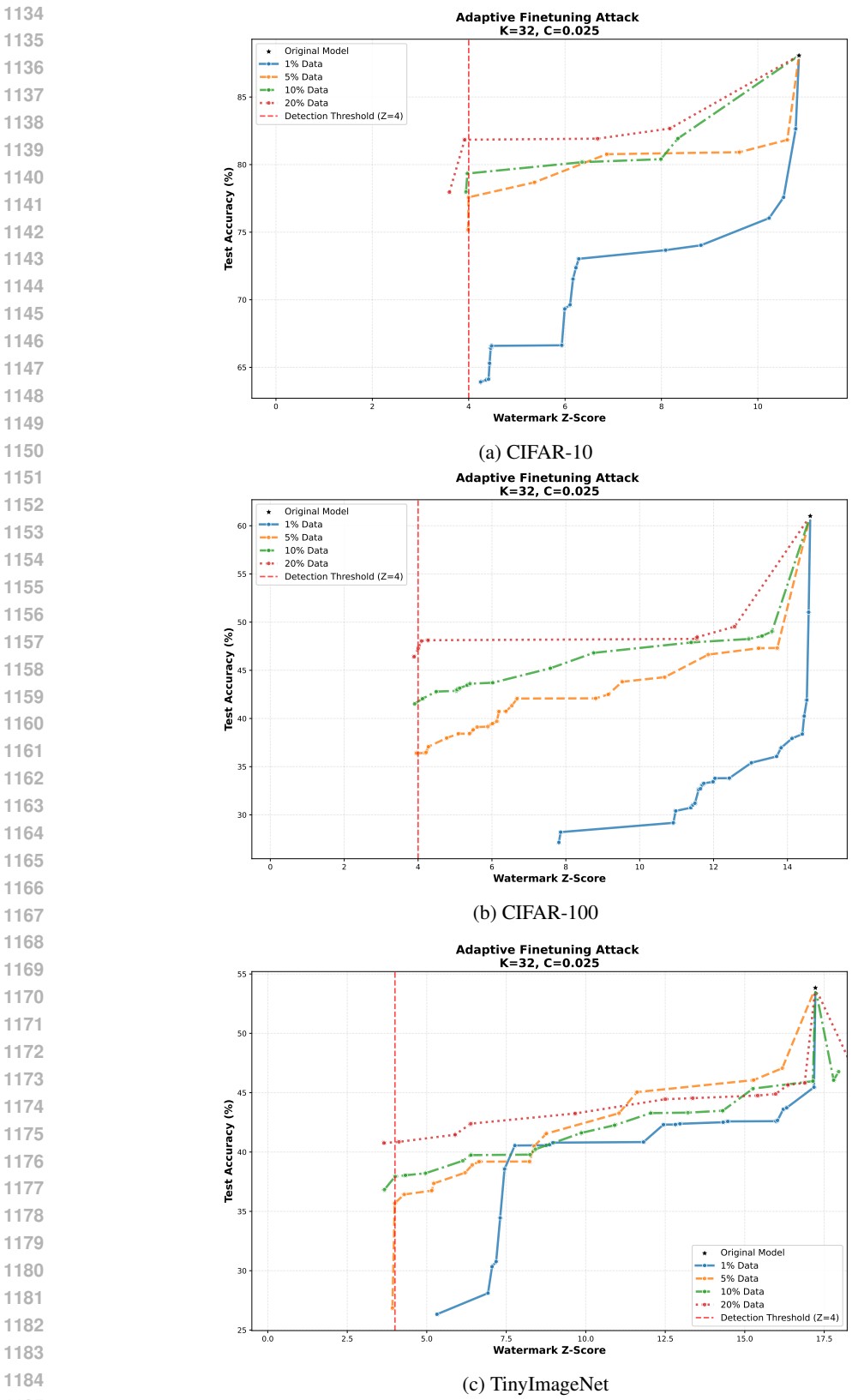

(a) CIFAR-10

(b) CIFAR-100

(c) TinyImageNet

Figure 7: Adaptive fine-tuning attacks with $K = 32$ clients and varying watermark strengths ($c \in \{0.075, 0.05, 0.025\}$). Each curve shows the trade-off between test accuracy and watermark detectability ($z$-score) as the adversary fine-tunes the watermarked model with different fractions of training data ($1\%, 5\%, 10\%, 20\%$). The red dashed line marks the detection threshold ($z^* = 4$).

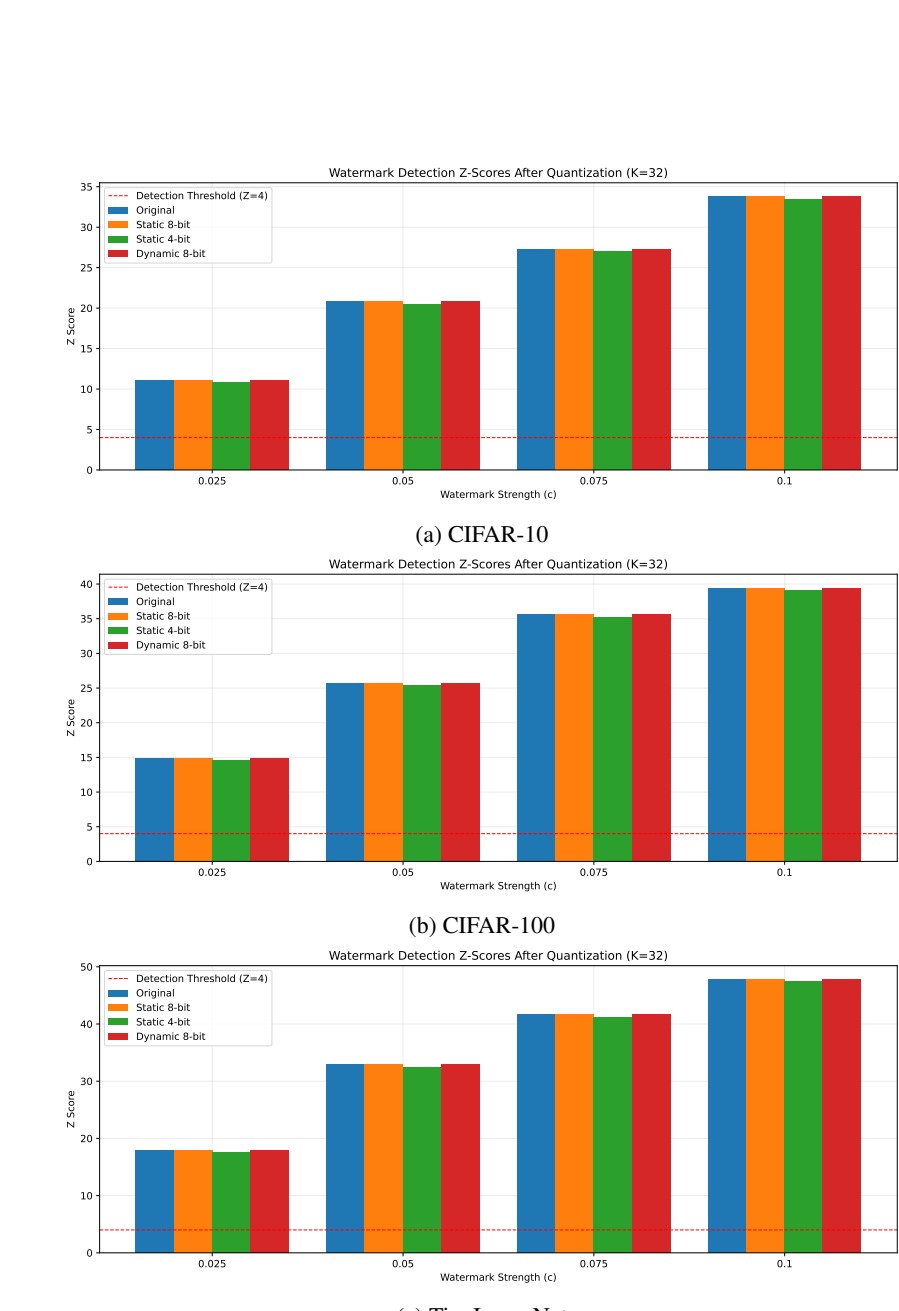

(a) CIFAR-10

(b) CIFAR-100

(c) TinyImageNet

Figure 8: Watermark detection $z$-scores after quantization. Results are shown for CIFAR-10 (a), CIFAR-100 (b), and TinyImageNet (c) under varying watermark strengths $c$. Each panel compares the original model with three quantization schemes. The dashed red line indicates the detection threshold ($z = 4$).

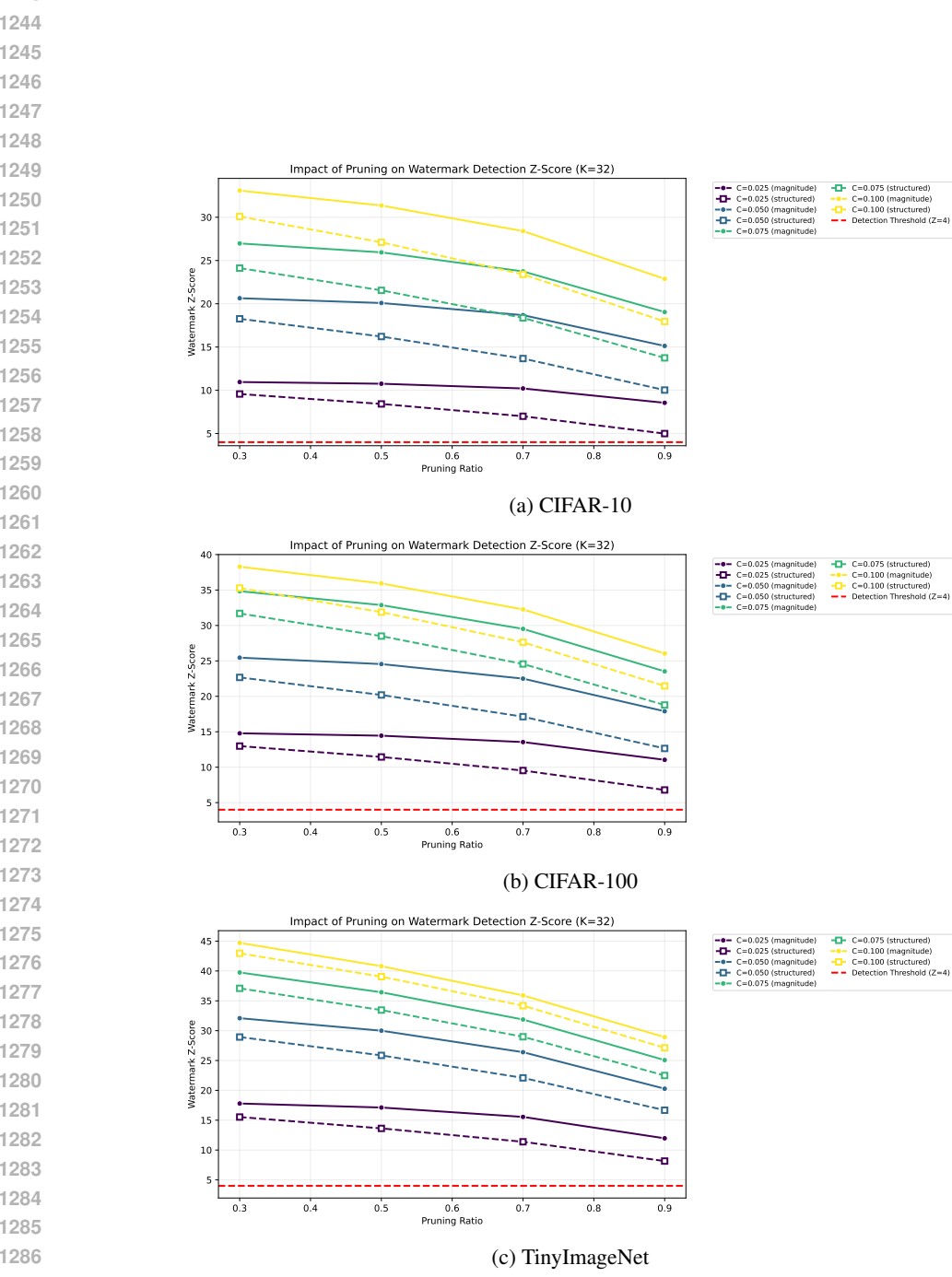

(a) CIFAR-10

(b) CIFAR-100

(c) TinyImageNet

Figure 9: Impact of pruning on watermark detection $z$-scores across datasets. Results are shown for (a) CIFAR-10, (b) CIFAR-100, and (c) TinyImageNet.

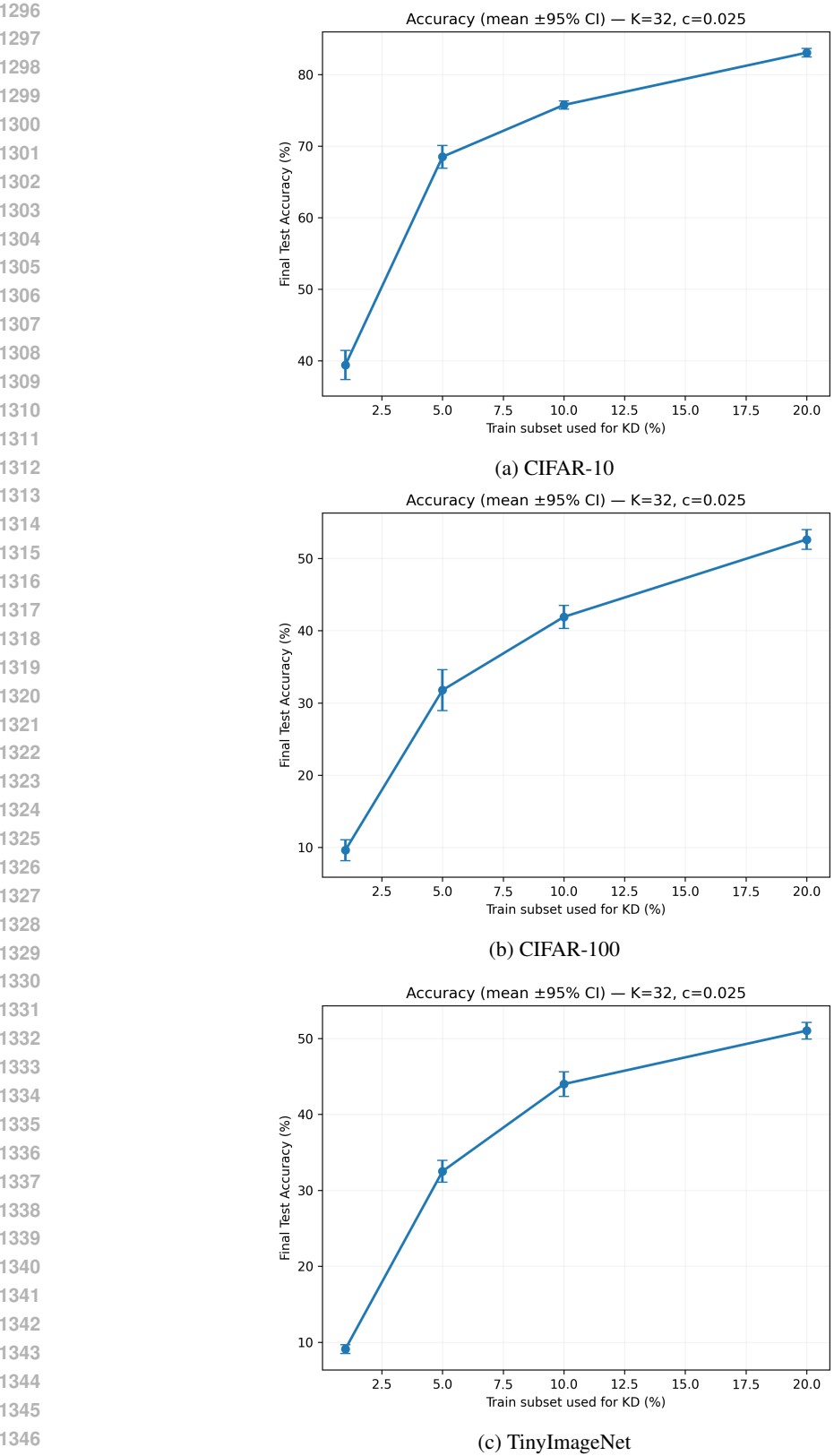

(a) CIFAR-10

(b) CIFAR-100

(c) TinyImageNet

Figure 10: Distillation attack with $K = 32$ clients and watermark strength $c = 0.025$. Final test accuracy is shown as the fraction of training data used for knowledge distillation increases. Results are reported for (a) CIFAR-10, (b) CIFAR-100, and (c) TinyImageNet. Error bars denote 95% confidence intervals across seeds.

