# OpenReview forum: "Collaborative Threshold Watermarking"
_ICLR.cc/2026/Conference — ICLR 2026 Conference Withdrawn Submission_

### Official Review · Reviewer_2ExH · 2025-10-26

**Soundness:** 3
**Presentation:** 3
**Contribution:** 3
**Rating:** 6
**Confidence:** 2

**Summary:**

The paper proposes a collaboratively verifiable watermark with shared embedding. It identifies that existing methods face the following issue: in federated learning, models jointly trained by multiple parties lack trust mechanisms and verifiability for ownership authentication. Existing watermarking methods assume a trusted server or a single authoritative party, which cannot ensure security and fairness in a "trustless multi-party" environment. To address this, the proposed method distributes watermark keys via secret sharing, and each client embeds watermark perturbations during training based on local gradient directions and EMA-based adaptive scaling. At least *t* clients must cooperate to recover the key and verify model ownership.

**Strengths:**

1. The authors provide a reasonable analysis of the limitations of existing methods: current approaches either assume a single trusted entity or fail to scale to large multi-client scenarios. Introducing "(t, K)-threshold watermarking" is a theoretical innovation that fills a gap in federated learning watermark mechanisms.
2. The use of cryptographic primitives (commitment schemes, secret sharing) ensures unforgeability and collusion resistance.
3. The experiments are comprehensive, and ablation studies are appropriately designed to demonstrate detectability, robustness, and model accuracy.

**Weaknesses:**

1. A main concern is the threat model: in which scenarios is it necessary to prove model ownership collaboratively by multiple parties under a white-box assumption? Furthermore, although the paper mentions a "trustless setting," potential attack scenarios (e.g., malicious clients tampering with gradients, server denial-of-service) are insufficiently discussed and are limited to an "honest-but-curious" model.
2. While the method is reasonable, the embedding perturbation direction relies on statistical assumptions in a high-dimensional parameter space (normality assumption) and lacks a theoretical error bound. The strategy for selecting the threshold *t* is not fully discussed, and the impact of different *t* values on communication, robustness, and security is unclear.
3. Experimentally, the datasets and models used are relatively small (e.g., ResNet-18). In practice, federated training typically involves larger models, and it remains to be verified whether the method’s performance scales to such settings.

**Questions:**

Please refer to weaknesses.

Overall, the paper proposes a novel watermarking scenario that could inspire future work in the field. However, it is recommended to provide examples illustrating potential real-world applications of the proposed threat model.

**Reviewer Familiarity**
While I am well-versed in model watermarking, my understanding of the goals and challenges of watermarking specifically in federated learning scenarios is limited.

---

### Official Review · Reviewer_7a97 · 2025-10-29

**Soundness:** 2
**Presentation:** 1
**Contribution:** 2
**Rating:** 2
**Confidence:** 3

**Summary:**

This paper introduces a (t, K)-threshold watermarking protocol for federated learning (FL), enabling only a coalition of at least t clients to verify a jointly embedded watermark, while smaller subsets learn nothing about it. The scheme combines commitment and secret-sharing primitives to provide collaborative, trustless ownership verification without a fully trusted server. Experiments on CIFAR-10, CIFAR-100, and TinyImageNet show that the watermark remains robust against several removal attacks (fine-tuning, pruning, quantization) while maintaining accuracy and scalability up to 128 clients.

**Strengths:**

- The paper introduces an interesting and timely idea of (t, K)-threshold watermarking for federated learning, addressing collaborative model ownership in untrusted multi-party settings.

- The formulation builds on established cryptographic primitives such as commitments and Shamir’s secret sharing, providing a reasonable conceptual foundation.

- The problem motivation is clear, and the overall goal of combining cryptographic guarantees with watermarking for provenance verification is relevant to the ICLR community.

- The empirical evaluation setup is broad in scope and includes multiple datasets and attack types, which, despite presentation flaws, indicates a substantial implementation effort.

**Weaknesses:**

- The presentation quality is very poor. Nearly all of the figures in both the main paper and the appendix have very small fonts and unreadable legends. The plots must be redrawn with larger, clearer labels and axes so they can be interpreted without zooming in 300%.  Moreover, the overall appearance of figures are not consistent, which does not meet ICLR’s standards. I strongly recommend that the authors fix this issue.

- The paper also uses the term “decrease by X%” when referring to accuracy drops (e.g., in Table 1). These are actually percentage-point decreases rather than relative percentages, and the phrasing should be corrected for clarity.

- The description of dataset complexity in lines 380–383 is confusing. TinyImageNet is actually more complex than CIFAR-10 and CIFAR-100, so the relative sensitivity statement should be revised or clarified.

- In Section 5.5, the authors list several watermark removal attacks (fine-tuning, adaptive fine-tuning, pruning, quantization, distillation) but do not cite any corresponding prior works or standard implementations in the main paper. Adding references to established attack schemes would both credit prior research and clarify how these attack variants were implemented.

- The discussion of robustness is relatively superficial. While several attacks are tested, the paper does not engage with the broader literature on watermark robustness and the ongoing attack–defense dynamics. Earlier works (Uchida et al., 2017; Zhang et al., 2018; Adi et al., 2018) initially claimed robustness against removal attacks, but subsequent studies (Shafieinejad et al., 2019; Aiken et al., 2020) demonstrated that such watermarks can in fact be removed through more sophisticated combinations of distillation, fine-tuning, and parameter regularization. Without referencing or discussing this line of work, the presented robustness claims risk appearing optimistic or attack-limited. A more detailed and reflective discussion that acknowledges this cat and mouse dynamic between watermarking and removal would considerably strengthen the paper and better situate it within the existing research landscape

- The lines 430-440 are somewhat unclear and grammatically inconsistent. In particular, the sentence describing when an attacker succeeds (“unless or...”) should be rewritten for clarity, and the explanation could be streamlined to avoid repetition. Apart from that, the referenced Figure 5 is not readable/interpretable as already mentioned above.

- While reading the paper I missed a proper related work section throughout. It only appears together with the discussion of results, which is unconventional and confusing. I recommend separating the two and providing a dedicated, well-written related work section earlier in the paper to establish context and prior work. In addition, the style of paragraph headings is inconsistent. Some end with “:”, some with “.”, and others without punctuation, further emphasizing a lack of editorial consistency.

- The focus on the white-box setting limits practical applicability, since most real-world verification scenarios are black-box. While the authors acknowledge this as a first step, it reduces impact.

**Questions:**

- Is Table 1 based on a single training seed per experiment? If yes, I recommend averaging results over multiple randomized seeds to make the reported accuracy drops and z-scores statistically meaningful.

- How is robustness defined quantitatively in your experiments? For instance, do you treat watermark detectability above z = 4 as a binary criterion, and if so, how was this threshold chosen or calibrated? Would alternative thresholds affect the reported robustness trends?

- What happens if fewer than t clients are available, either during training or verification?
Can the scheme still function correctly, or is watermark embedding or verification delayed until at least t clients are active?

- How sensitive is the verification threshold (z ≥ 4) to model architecture and training noise? Have you tested it beyond ResNet-18?

- The related work section is not sufficiently detailed to understand how this approach compares to existing FL watermarking methods. Could the authors clarify which other federated watermarking schemes are most relevant and why direct quantitative comparisons were not included?

---

### Official Review · Reviewer_zYaE · 2025-10-31

**Soundness:** 2
**Presentation:** 1
**Contribution:** 2
**Rating:** 2
**Confidence:** 4

**Summary:**

In this paper, the authors propose $(t,k)$-threshold watermarking for federated learning, which allows multiple clients (the minimum required number being $t$) to jointly verify the presence of the watermark for detecting unauthorized use of the model or claiming model ownership. The method combines secret sharing, cryptographic commitments, and secure aggregation to enable a collaborative watermark that remains verifiable in a white-box setting. The authors provide formal definitions and empirical evaluations for CIFAR-10, CIFAR-100, and TinyImageNet. $(t,k)$-threshold watermarking is shown to be robust against different watermark removal attacks while the degradation in model utility is minimal.

The claimed contribution is conceptually interesting. However, for acceptance, the threat model must be clearly formalized, the evaluation must strengthen the theoretical claims, and there must be a comparison to existing FL watermarking frameworks.

**Strengths:**

S1.  Novel Problem Formulation: The authors propose a new type of watermarking method suitable for federated learning, where the verification of the model is carried out collectively by a subset of clients and requires $\geq t$ participants for verification.

S2. Cryptographic Grounding: The scheme is built on standard cryptographic primitives (commitment, secret sharing, and secure aggregation).

**Weaknesses:**

W1. Lack of Clarity in Threat Model: The threat model is vague (who is the adversary, what is their goal, who is the trusted dealer, do you have a trusted or trustless server?) and conflicts with the scenarios. (Please check the first two questions for a detailed comments)

W2. Underexplored Practical Deployment Considerations: The scheme assumes an initial trusted dealer for share distribution. How this dealer is instantiated in real federated settings is left unaddressed.

W3. Limited Empirical Guarantees: The authors describe important properties such as correctness, soundness, and threshold security, but do not provide empirical evidence to strengthen these theoretical claims.

W2. White-box Verification: The verification process requires white-box access. This is not a practical scenario if a client is the malicious adversary who distributes the model in an authorized way. This is also related to the first weakness, who is the adversay and what is the main goal that necessitates the white-box verification?

**Questions:**

Q1. Who is the trusted dealer? Can you give a use case or example to better explain this role? Who is the adversary, a malicious client or an end user?

Q2. In the threat model, the authors mention that there is no trusted server. However, there is a server responsible for secure aggregation, but cannot be trusted. Later, they also present a scenario; where the server is trusted. I suggest authors to reword or remove this sentence, as the current one is perceived as completely decentralized (peer-to-peer) federated learning. I also suggest authors to carefully revise their threat model, since the current text conflicts with the variants of their protocol and is vague in definitions.

Q3. The communication/computation overhead of secret sharing and reconstruction is not quantified.

Q4. The proposed scheme is highly dependent on the threshold t and the preset $z^*$. There is no empirical analysis of how these affect the watermarking embedding and verification.

Q5. Abstract lacks the federated learning term. Only distributed settings are mentioned, which covers many different settings rather than federated learning only.

Q6. Figure 5 is very hard to read.

Q7. There are typos and incorrect in-text references in the text in Section 6. Section 6 also lacks discussion even if the title says discussion.

Q8. References on page 13 do not fit to the format (too many spaces between each reference).

**Details Of Ethics Concerns:**

No ethics concerns

---

### Official Review · Reviewer_4S6K · 2025-10-31

**Soundness:** 2
**Presentation:** 2
**Contribution:** 2
**Rating:** 2
**Confidence:** 4

**Summary:**

This paper proposes a thresholded watermarking scheme for models trained with Federated Learning. In more detail, the method embeds a signal in the weights of the model which can be used to verify model provenance but is infeasible to detect without knowledge of some hidden information -- this is the general idea of model watermarking, which is explored in previous work. The contribution of this paper is that they design a watermarking scheme that cannot be verified without agreement from a threshold number of the FL clients.

**Strengths:**

The writing is stylistically good -- in general the paper is communicated clearly.

The proposed primitive of a threshold verifiable watermark is novel, and abstractly interesting.

**Weaknesses:**

- **trusted server setting makes motivation for threshold security much weaker** --  if there is a trusted server which knows $\tau$, I'm not sure I understand why distributed threshold verification by clients is needed. If the server is trusted, then surely they can be trusted to compute the verification on the behalf of the clients as well. Are there any applications where this would not be the case?

- **'trustless server' setting requires a trusted server** -- I don't think the 'trustless server' setting in the paper is named appropriately. It requires a trusted dealer to distribute additive shares, and specifies that the dealer is not one of the clients. So what can the trusted dealer be called other than a trusted server? To me this detracts substantially from the motivation of the work -- simply requiring a *different* trusted server to create $\tau$ and hide it from the aggregation server has a lot of the same drawbacks as the initial trusted server setting. This should be mentioned in the limitations section, and authors should make an argument for why a trusted dealer is justifiable. This limitation could potentially be addressed by using secure multiparty computation to perform the work of the dealer (see e.g. [1]).

- **fit of semi-honest security model for clients in FL; additive shares are brittle to ubiquitous issues in real-world FL execution (even when semi-honest assumption holds)** -- In most common FL settings there is very little to prevent clients from performing malicious behavior. Unless I'm misunderstanding something, it appears that the aggregation in line 37 of Alg 1 is very brittle to even a single Byzantine client (see [2] if unfamiliar). Further, even if all clients adhere to the semi-honest assumption, it appears to me that the aggregation is also extremely brittle to *unintentional* perturbations that are ubiquitous to FL such as client dropout -- e.g. if even one client fails to submit their update, this destroys the integrity of the additively shared value, replacing it with a uniform random value.

- **(at least some of) the protocol should be in main text** -- relegating Algorithm 1 to the appendix delays answers to a lot of natural questions that crop up while reading the paper. Further, it could obscure a lot of the limitations mentioned above from a cursory reader. I understand that Alg 1 takes up a lot of space and the page limit is tight, but I think the exposition would benefit a lot from having it in the main text. I think you could potentially save space by dividing it into a few subroutines and displaying them in two-column format. Maybe some of the less important subroutines could be relegated to the appendix (in my opinion, the trusted server variant is a good candidate -- the 'trustless' setting is the main contribution).

- **needs more concrete motivation for threshold watermark verification** -- the motivation given for why threshold verification is important is a bit abstract. I think the paper would benefit substantially if the authors described a use case where the primitive addresses a concrete problem. Of note, the third paragraph of the introduction is a strong argument for why *FL watermarking* is necessary, but it's unclear why *thresholded* watermarking is specifically important to address the challenges raised. When I first read this paragraph, it gave me the impression that the authors were the first to extend watermarking to the FL setting, but later the related work revealed that this is not the case.

[1] Robust and Actively Secure Serverless Collaborative Learning. Franzese et al. 2023

[2] Machine Learning with Adversaries: Byzantine Tolerant Gradient Descent. Blanchard et al. 2017

**Questions:**

My questions/suggestions are mostly articulated in the weaknesses, but I will reiterate/condense them here:

- In the trusted server setting, are there any applications where it makes more sense to use threshold verification rather than direct verification by the trusted server?
- The assumption of a trusted dealer in the 'trustless server' setting should be mentioned more prominently, and authors should make an argument for why it is justified. Consider using secure multiparty computation to remove this requirement.
- Am I correct in my understanding that dropout or adversarial perturbation of updates from even a single client is highly problematic for model integrity in the 'trustless server' setting? If not, can you demonstrate why it's not as problematic as it appears?
- Revise so that (at least the most important parts of) Algorithm 1 are in the main text.
- Can you add a more concrete use case for threshold watermark verification?

---

### Note · Authors · 2025-11-24

**Comment:**

We’d like to withdraw our paper from ICLR. We truly appreciate the reviewers’ thoughtful comments and the effort that went into the review process! We’ve received valuable feedback and plan to build on it for a future version of the work.

**Withdrawal Confirmation:**

I have read and agree with the venue's withdrawal policy on behalf of myself and my co-authors.